# Neural Speed Reading with Structural-Jump-LSTM

**Christian Hansen, Casper Hansen, Stephen Alstrup, Jakob Grue Simonsen & Christina Lioma**
Department of Computer Science
University of Copenhagen
Denmark, Copenhagen 2100
`{chrh,c.hansen,s.alstrup,simonsen,c.lioma}@di.ku.dk`

## Abstract

Recurrent neural networks (RNNs) can model natural language by sequentially "reading" input tokens and outputting a distributed representation of each token. Due to the sequential nature of RNNs, inference time is linearly dependent on the input length, and all inputs are read regardless of their importance. Efforts to speed up this inference, known as "neural speed reading", either ignore or skim over part of the input. We present Structural-Jump-LSTM: the first neural speed reading model to both skip and jump text during inference. The model consists of a standard LSTM and two agents: one capable of skipping single words when reading, and one capable of exploiting punctuation structure (sub-sentence separators (,:), sentence end symbols (.!?), or end of text markers) to jump ahead after reading a word. A comprehensive experimental evaluation of our model against all five state-of-the-art neural reading models shows that Structural-Jump-LSTM achieves the best overall floating point operations (FLOP) reduction (hence is faster), while keeping the same accuracy or even improving it compared to a vanilla LSTM that reads the whole text.

## 1 Introduction

Recurrent neural networks (RNNs) are a popular model for processing sequential data. The Gated Recurrent Unit (GRU) (Chung et al., 2014) and Long Short Term Memory (LSTM) (Hochreiter & Schmidhuber, 1997) are RNN units developed for learning long term dependencies by reducing the problem of vanishing gradients during training. However, both GRU and LSTM incur fairly expensive computational costs, with e.g. LSTM requiring the computation of 4 fully connected layers for each input it reads, independently of the input's importance for the overall task.

Based on the idea that not all inputs are equally important, and that relevant information can be spread throughout the input sequence, attention mechanisms were developed (Bahdanau et al., 2015) to help the network focus on important parts of the input. With *soft* attention, all inputs are read, but the attention mechanism is fully differentiable. In comparison, *hard* attention completely ignores part of the input sequence. Hard attention mechanisms have been considered in many areas, ranging from computer vision (Mnih et al., 2014; Campos et al., 2018) where the model learns what parts of the image it should focus on, to natural language processing (NLP), such as text classification and question answering (Yu et al., 2017; Campos et al., 2018; Yu et al., 2018), where the model learns which part of a document it can ignore. With hard attention, the RNN has fewer state updates, and therefore fewer floating point operations (FLOPs) are needed for inference. This is often denoted as *speed reading*: obtaining the same accuracy while using (far) fewer FLOPs (Yu et al., 2017; 2018; Seo et al., 2018; Huang et al., 2017; Fu & Ma, 2018). Prior work on speed reading processes text as chunks of either individual words or blocks of contiguous words. If the chunk being read is important enough, a full state update is performed; if not, the chunk is either ignored or a very limited amount of computations are done. This is followed by an action aiming to speed up the reading, e.g. skipping or jumping forward in text.

Inspired by human speed reading, we **contribute** an RNN speed reading model that ignores unimportant words in important sections, while also being able to jump past unimportant sections of the

text. Our model, called Structural-Jump-LSTM[1], **both** skips and jumps over dynamically defined chunks of text as follows: (a) it can skip individual words, after reading them, but before updating the RNN state; (b) it uses the punctuation structure of the text to define dynamically spaced jumps to the next sub-sentence separator (,;), end of sentence symbol (.!?), or the end of the text.

An extensive experimental evaluation against *all* state-of-the-art speed reading models (Seo et al., 2018; Yu et al., 2017; 2018; Fu & Ma, 2018; Huang et al., 2017), shows that our Structural-Jump-LSTM of dynamically spaced jumps and word level skipping leads to large FLOP reductions while maintaining the same or better reading accuracy than a vanilla LSTM that reads the full text.

## 2 RELATED WORK

Prior work on speed reading can be broadly clustered into two groups: 1) jumping based models, and 2) word level skip and skim models, which we outline below.

**Jump based models.** The method of Yu et al. (2017) reads a fixed number of words, and then may decide to jump a varying number of words ahead (bounded by a maximum allowed amount) in the text, or to jump directly to the end. The model uses a fixed number of total allowed jumps, and the task for the network is therefore to learn how best to spend this *jump budget*. The decision is trained using reinforcement learning, with the REINFORCE algorithm (Williams, 1992), where the reward is defined based only on if the model predicts correctly or not. Thus, the reward does not reflect how much the model has read. The model of Fu & Ma (2018) also has a fixed number of total jumps and is very similar to the work by Yu et al. (2017), however it allows the model to jump both back and forth a varying number of words in order to allow for re-reading important parts. Yu et al. (2018) use a CNN-RNN network where a block of words is first read by a CNN and then read as a single input by the RNN. After each block is read, the network decides to either re-read the block, jump a varying number of blocks ahead, or jump to the end. The decision is trained using reinforcement learning, where both REINFORCE and actor-critic methods were tested, with the actor-critic method leading to more stable training. The reward is based on the loss for the prediction and the FLOPs used by the network to make the prediction. FLOP reduction is thereby directly tied into the reward signal. Huang et al. (2017) propose a simple early-stopping method that uses a RNN and reads on a word level, and where the network learns when to stop. This can be considered a single large jump to the end of the text.

**Skip and skim based models.** Seo et al. (2018) present a model with two RNNs, a "small" RNN and a "big" RNN. At each time step the model chooses either the big or the small RNN to update the state, based on the input and previous state, which can be considered as text skimming when the small RNN is chosen. This network uses a Gumbel softmax to handle the non-differentiable choice, instead of the more common REINFORCE algorithm. Campos et al. (2018) train a LSTM that may choose to ignore a state update, based on the input. This can be considered as completely skipping a word, and is in contrast to skimming a word as done by Seo et al. (2018). This network uses a straight-through estimator to handle the non-differentiable action choice. This approach is applied on image classification, but we include it in this overview for completeness.

**Other speed reading models**. Johansen & Socher (2017) introduce a speed reading model for sentiment classification where a simple model with low computational cost first determines if an LSTM model should be used, or a bag-of-words approach is sufficient. The method of Choi et al. (2017) performs question answering by first using a CNN-based sentence classifier to find candidate sentences, thereby making a summary of the whole document relevant for the given query, and then using the summary in an RNN.

More widely, gated units, such as GRU and LSTM, face problems with long input sequences (Neil et al., 2016). Speed reading is one way of handling this problem by reducing the input sequence. In contrast, Neil et al. (2016) handle the problem by only allowing updates to part of the LSTM at a current time point, where an oscillating function controls what part of the LSTM state can currently be updated. Cheng et al. (2016) handle this by using memory networks to store an array of states, and the state at a given point in time comes from applying an attention mechanism over the stored states, handling the issues of older states being written over.

---

[1] https://github.com/Varyn/Neural-Speed-Reading-with-Structural-Jump-LSTM

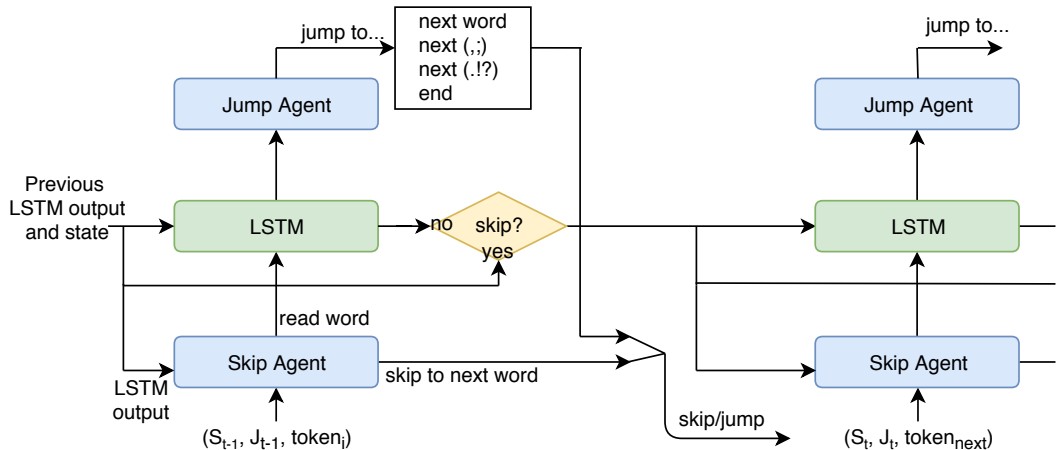

Figure 1: Overview of our proposed model. The input at a given time is the action of the previous skip agent ($S_{t-1}$), the previous jump agent action ($J_{t-1}$), and the word embedded token (token$_i$). token$_{next}$ corresponds to the next word considered after skipping or jumping. Depending on the skip decision, the no/yes in the diamond shaped skip-box corresponds to which LSTM output and state should be used for the next input (updated or previous respectively).

Overall, the above state-of-the-art models are either jump or skip/skim based. We present the first speed reading model that both jumps and skips. Furthermore, current jumping-based models use a variable jump size for each input, without considering the inherent structure of the text. In contrast, our model defines jumps based on the punctuation structure of the text. This combined approach of both skipping and jumping according to text structure yields notable gains in efficiency (reduced FLOPs) without loss of effectiveness (accuracy). We next present our model.

## 3  STRUCTURAL-JUMP-LSTM MODEL

Our Structural-Jump-LSTM model consists of an ordinary RNN with LSTM units and two agents: the *skip* agent and the *jump* agent. Each of these agents compute a corresponding action distribution, where the skip and jump actions are sampled from. The skip agent chooses to either skip a single word, thereby not updating the LSTM, or let the LSTM read the word leading to an update of the LSTM. The jump agent is responsible for jumping forward in the text based on punctuation structure (henceforth referred to as *structural* jumps). A structural jump is a jump to the next word, or the next sub-sentence separator symbol (,;), or the next end of sentence symbol (.!?), or to the end of the text (which is also an instance of end of sentence). The purpose of using two agents is that the skip agent can ignore unimportant words with very little computational overhead, while the jump agent can jump past an unimportant part of the text. As both the skip and jump agent contribute to a reduction in FLOPs (by avoiding LSTM state updates), the Structural-Jump-LSTM is faster at inference than a vanilla LSTM.

Figure 1 shows an overview of our model: The input in each time step is the previous actions of the skip agent (S), of the jump agent (J), and of the current input. The output from the previous LSTM is used in combination with the input to make a *skip decision* – if the word is skipped, the last LSTM state will not be changed. From this we use a standard LSTM cell where the output is fed to the jump agent, and a *jump decision* is made. Both agents make their choice using a fully connected layer, with a size that is significantly smaller than the LSTM cell size, to reduce the number of FLOPs by making the overhead of the agents as small as possible.

Section 3.1 details how inference is done in this model, and Section 3.2 presents how the network is trained.

## 3.1 INFERENCE

At a given time step $t$, Structural-Jump-LSTM reads input $x_i \in \mathbb{R}^d$ , and the LSTM has a previous output $o_{t-1} \in \mathbb{R}^m$ and state $s_{t-1} \in \mathbb{R}^m$. At time step $t-1$ the skip agent first takes action $a_{t-1}^{\text{skip}}$ sampled from the skip-action distribution $p_{t-1}^{\text{skip}}$ and the jump agent takes action $a_{t-1}^{\text{jump}}$ sampled from the jump-action distribution $p_{t-1}^{\text{jump}}$. If $a_{t-1}^{\text{skip}}$ is to skip the word, then the jump agent takes no action, i.e. no jump is made. At time step $t$ the network first needs to sample $a_t^{\text{skip}}$ from $p_t^{\text{skip}}$, which is computed in each time step as:

$$p_t^{\text{skip}} = \text{softmax}(d_{\text{LIN}}(\text{state}_t^{\text{skip}})) \tag{1}$$

$$\text{state}_t^{\text{skip}} = d_{\text{ReLU}}(x_t \odot o_{t-1} \odot \text{onehot}(a_{t-1}^{\text{skip}}) \odot \text{onehot}(a_{t-1}^{\text{jump}})) \tag{2}$$

where $d_{activation}$ is a fully connected layer with the given activation, ReLU is the Rectified Linear Unit, LIN is the linear activation, and $\odot$ denotes concatenation of vectors. At inference time the action can either be sampled from the distribution, or chosen greedily, by always choosing the most probable action.

If the action $a_t^{\text{skip}} = 0$, we skip the word and set $o_t = o_{t-1}$ and $s_t = s_{t-1}$, and the network will move to the next word at position $i + 1$. If $a_t^{\text{skip}} = 1$, the word is read via the LSTM. The output and new state of the RNN is calculated and produces $o_t$ and $s_t$ for the next step. The probability distribution $p_t^{\text{jump}}$ from which action $a_t^{\text{jump}}$ is sampled from is computed as:

$$p_t^{\text{jump}} = \text{softmax}(d_{\text{LIN}}(\text{state}_t^{\text{jump}})) \tag{3}$$

$$\text{state}_t^{\text{jump}} = d_{\text{ReLU}}(o_t) \tag{4}$$

If the sampled action $a_t^{\text{jump}}$ corresponds to e.g. a jump to the next sentence, then the current LSTM output and state will be kept, and all following inputs will be ignored until a new sentence begins. When there are no more inputs the output of the RNN is used to make a final prediction. If the action is to jump to the end of the text, then the final prediction will be made immediately based on the current output.

## 3.2 TRAINING

During training, Structural-Jump-LSTM is optimized with regards to two different objectives: 1) Producing an output that can be used for classification, and 2) learning when to skip and jump based on the inputs and their context, such that the minimum number of read inputs gives the maximum accuracy.

For objective 1, the output of the RNN can be used for classification, and the loss is computed as the cross entropy $L_{\text{class}}$ against the target.

For objective 2, the agents are non-differentiable due to the discrete sampling of the actions. Thus we choose to reformulate the problem as a reinforcement learning problem, where we define a reward function to maximize. In essence, the reward is given based on the amount read and whether or not the prediction is correct. We denote $R$ as the total reward associated with a sampled sequence of actions, e.g. $a_1^{\text{skip}}, a_2^{\text{skip}} a_2^{\text{jump}}, ..., a_T^{\text{skip}}, a_T^{\text{jump}}$, and $R_t$ as the sum of reward from time $t$. Note that if the network chooses to skip a word, the sequence will not have a jump at that time step. We use an advantage actor-critic approach (Konda & Tsitsiklis, 2000) to train the agents in order to reduce the variance. The loss for the skip agent is given as:

$$L_{\text{actor}} = -\sum_{t=0}^{T} \log(p_t^{\text{skip}}(a_t^{\text{skip}}|\text{state}_t^{\text{skip}})) \cdot (R_t - V_t^{\text{skip}}) \tag{5}$$

$$V_t^{\text{skip}} = d_{\text{LIN}}(\text{state}_t^{\text{skip}}) \tag{6}$$

where $V_t^{\text{skip}}$ is a value estimate of the given state, which is produced by a fully connected layer with output size 1. For the jump agent we do exactly the same as the skip agent, and the sum of the two actor losses is denoted $L_{\text{actors}}$. The value estimate of a state corresponds to how much reward is

| dataset | task type | label type | #train | #val | #test | avg. length | vocab. size |
|---|---|---|---|---|---|---|---|
| IMDB (Maas et al., 2011) | Sentiment | Pos/Neg | 21,250 | 3,750 | 25,000 | 230 | 204,758 |
| Yelp (Zhang et al., 2015) | Sentiment | Pos/Neg | 475,999 | 84,000 | 37,999 | 136 | 644,653 |
| SST (Socher et al., 2013) | Sentiment | Pos/Neg | 6,920 | 872 | 1,821 | 19 | 17,851 |
| Rotten Tomatoes (Pang & Lee, 2005) | Sentiment | Pos/Neg | 9,594 | 1,068 | 1,068 | 21 | 20,388 |
| DBPedia (Lehmann et al., 2015) | Topic | 14 topics | 475,999 | 84,000 | 69,999 | 47 | 840,843 |
| AG news (Zhang et al., 2015) | Topic | 4 topics | 101,999 | 18,000 | 7,599 | 8 | 41,903 |
| CBT-CN (Hill et al., 2016) | Q/A | 10 answers | 120,769 | 2,000 | 2,500 | 429 | 51,774 |
| CBT-NE (Hill et al., 2016) | Q/A | 10 answers | 108,719 | 2,000 | 2,500 | 394 | 51,672 |

Table 1: Dataset statistics.

collected when acting from this state, such that the estimated value is a smoothed function of how much reward the network expects to collect later. Using advantage instead of reward is beneficial as the sign of the loss then depends on whether the achieved reward is higher or lower than the expected reward in a given state. This loss for the agent corresponds to the loss in the popular A3C algorithm (Mnih et al., 2016) in the case of $t_{\max}$ being large enough to always reach a terminal condition. This is not a problem in our setting as documents are of finite length. The value estimate is trained together with both agents using the squared difference with the targets being the observed values for each state, (we denote this $L_{\text{critics}}$). Lastly, to provoke exploration in the network we add an entropy loss, where both agents' distributions are targeting a uniform distribution over the actions, (we denote this loss $L_{\text{entropies}}$). The total loss for the network is then:

$$L_{\text{total}} = \alpha L_{\text{class}} + \beta L_{\text{actors}} + \gamma L_{\text{critics}} + \delta L_{\text{entropies}} \tag{7}$$

where $\alpha, \beta, \gamma$, and $\delta$ control the trade-offs between the components.

For each action a reward is given; the reward for a skip action at time $t$ is:

$$r_t^{\text{skip}} = \begin{cases} -\frac{1}{|\text{doc}|} & \text{if } a_t^{\text{skip}} \text{ is a read action} \\ -\frac{c_{\text{skip}}}{|\text{doc}|} & \text{if } a_t^{\text{skip}} \text{ is a skip action} \end{cases} \tag{8}$$

where $|\text{doc}|$ is the number of words in the document such that the reward for skipping a word scales with the document length. The reward is negative for both cases as there is a cost associated with reading a word. The jump action gives no reward, as the reward is implicit by the network collecting less negative reward. At the end an additional reward is given based on whether the network makes a correct prediction, such that the summed reward from time $t$ is given by:

$$R_t = \begin{cases} 1 + w_{\text{rolling}} \sum_{t'=t}^{T} r_{t'}^{\text{skip}} & \text{if } y_{\text{pred}} = y_{\text{target}} \\ p(y_{\text{target}}) + w_{\text{rolling}} \sum_{t'=t}^{T} r_{t'}^{\text{skip}} & \text{if } y_{\text{pred}} \neq y_{\text{true}} \end{cases} \tag{9}$$

$y_{\text{pred}}$ is the prediction by the network, $y_{\text{true}}$ is the target, and $p(y_{\text{target}})$ is the probability the network gives for the target class. $w_{\text{rolling}}$ controls the trade-off between the rolling reward and the reward based on model performance. The final reward is designed such that a large reward is given in the case of a correct prediction, while the agents are still rewarded for increasing the probability for the correct class, even if they did not predict correctly.

## 4 EXPERIMENTAL EVALUATION

We present the experimental evaluation of our method.

### 4.1 EXPERIMENTAL SETUP AND TRAINING

We use the same tasks and datasets used by the state-of-the-art in speed reading (displayed in Table 1), and evaluate against all 5 state-of-the-art models (Seo et al., 2018; Yu et al., 2017; 2018; Fu & Ma, 2018; Huang et al., 2017) in addition to a vanilla LSTM full reading baseline.

For the sentiment and topic classification datasets we apply a fully connected layer on the LSTM output followed by a traditional softmax prediction, where the fully connected layer has the same size as the cell size. On the Question Answering datasets we follow Yu et al. (2017) by choosing the candidate answer with the index that maximizes $\text{softmax}(CWo) \in \mathbb{R}^{10}$, where $C \in \mathbb{R}^{10 \times d}$ is the word embedding matrix of the candidate answers, $d$ is the embedding size, $W \in \mathbb{R}^{d \times \text{cell\_size}}$ is a

trained weight matrix, and $o$ is the output state of the LSTM. This transforms the answering task into a classification problem with 10 classes. In addition, we read the query followed by the document, to condition reading of the document by the query as done by Yu et al. (2017). On all datasets we initialize the word embedding with GloVe embeddings (Pennington et al., 2014), and use those as the input to the skip agent and LSTM.

We use the predefined train, validation, and testing splits for IMDB, SST, CBT-CN, and CBT-NE, and use 15% of the training data in the rest as validation. For Rotten Tomatoes there is no predefined split, so we set aside 10% for testing as done by Yu et al. (2017). For training the model we use RM-Sprop with a learning rate chosen from the set $\{0.001, 0.0005\}$, with optimal of 0.001 on question answering datasets (CBT-CN and CBT-NE) and 0.0005 on the topic and sentiment datasets. We use a batch size of 32 on AG news, Rotten Tomatoes, and SST and a batch size of 100 for the remaining. Similarly to Yu et al. (2017), we employ dropout to reduce overfitting, with 0.1 on the embedding and 0.1 on the output of the LSTM. For RNN we use an LSTM cell with a size of 128, and apply gradient clipping with a tresholded value of 0.1. For both agents, their small fully connected layer is fixed to 25 neurons.

On all datasets we train by first maximizing the full read accuracy on the validation set, and the agents are then activated afterwards. While training we include the entropy loss in the total loss to predispose the speed reading to start with a full read behaviour, where the action distributions are initialized to only reading and never skipping or jumping. While maximizing full read accuracy the word embedding is fixed for the Question Answering datasets and trainable on the rest, while being fixed for all datasets during speed read training. As described in Equation 9, $w_{\text{rolling}}$ controls the trade-off between correct prediction and speed reading, and was chosen via cross validation from the set $\{0.05, 0.1, 0.15\}$, where most datasets performed best with 0.1. For simplicity we fix the cost of skipping $c_{\text{skip}}$ in Equation 8 to 0.5, such that skipping a word costs half of reading a word, which was done to promote jumping behaviour.

For the speed reading phase the total loss, as seen in Equation 7, is a combination of the prediction loss, actor loss, critic loss, and entropy loss, where the actor loss is scaled by a factor of 10 to make it comparable in size to the other losses. The entropy loss controls the amount of exploration and is chosen via cross validation from the set $\{0.01, 0.05, 0.1, 0.15\}$, where most datasets performed best with 0.1. We also cross validate choosing the actions greedily or via sampling from the action distributions, where for QA sampling was optimal and greedy was optimal for the others. Lastly, all non-QA datasets use uniform action target distributions for increased exploration, however CBT-CB and CBT-CN are trained with distribution with 95% probability mass on the "read" choice of both agents to lower skipping and jumping exploration, which was necessary to stabilize the training.

## 4.2 EVALUATION METRICS

The objective of speed reading consists of two opposing forces, as the accuracy of the model should be maximized while reading as few words as possible. We consider two different ways the model can skip a word: i) One or more words can be jumped over, e.g. as done in our model, LSTM-Jump (Yu et al., 2017), Yu-LSTM (Yu et al., 2018) and Adaptive-LSTM (Huang et al., 2017), where the latter implements a jump as early stopping. ii) A word can be skipped or skimmed, where the model is aware of the word (in contrast to jumping), but chooses to do a very limited amount of computations based on it, e.g. as done as skipping in our model or as skimming in Skim-LSTM (Seo et al., 2018). In order to capture both of these speed reading aspects, we report the percentage of words jumped over, and the total reading percentage (when excluding skipped and jumped words).

We calculate the total FLOPs used by the models as done by Seo et al. (2018) and Yu et al. (2018), reported as a FLOP reduction (FLOP-r) between the full read and speed read model. This is done to avoid runtime dependencies on optimized implementations, hardware setups, and whether the model is evaluated on CPU or GPU.

## 4.3 RESULTS

We now present the results of our evaluation.

> (Class: Mean Of transportation) The ~~Alexander Dennis~~ Enviro200 Dart ~~is~~ a midibus manu-factured by ~~Alexander Dennis~~ since 2006 ~~for the British~~ market as ~~the~~ successor ~~of the~~ Dart ~~SLF chassis and Pointer body.~~ ~~The~~ Enviro200 ~~Dart is manufactured and marketed in North~~ ~~America by New Flyer as the MiDi~~.

Figure 2: Example of jumping and skipping behaviour of our Structural-Jump-LSTM from DB-Pedia. Skipped words have a single strike-through while jumps consist of a sequence of strike-throughed words. The words that are jumped over or skipped are considered by the model not important for classifying the means of transportation (even though they include several nouns and name entinites that are generally considered to be important (Lioma & van Rijsbergen, 2008).

### 4.3.1 STRUCTURAL-JUMP-LSTM VERSUS FULL READING

Table 2 displays the accuracy, the percentage of text being jumped over, and the total reading per-centage (when excluding jumped and skipped words), of our approach versus a full reading baseline. Our approach obtains similar accuracies compared to vanilla LSTM, while in some cases reducing the reading percentage to below 20% (IMDB and DBPedia), and with the worst speed reading result-ing in a reading percentage of 68.6% (Yelp). The speed reading behaviour varies across the datasets with no skipping on CBT-CN, CBT-NE, and Yelp, no jumping on Rotten Tomatoes, and a mix of skipping and jumping on the remaining datasets.

In 7 out of 8 datasets Structural-Jump-LSTM improves accuracy and in 1 the accuracy is the same (IMDB). At all times, our model reads significantly less text than the full reading baseline, namely 17.5% - 68.8% of the text. Overall this indicates that the jumps/skips of our model are meaningful (if important text was skipped or jumped over, accuracy would drop significantly). An example of this speed reading behaviour (from DBPedia) can be seen in Figure 2. Generally, the model learns to skip some uninformative words, read those it deems important for predicting the target (Mean of transportation), and once it is certain of the prediction it starts jumping to the end of the sentences. Interestingly, in this setting it does not learn to just jump to the end, but rather to inspect the first two words of the last sentence.

### 4.3.2 STRUCTURAL-JUMP-LSTM VERSUS STATE-OF-THE-ART SPEED READING

Table 3 displays the scores of our approach against all five state-of-the-art speed reading models. We report the values from the original papers, which all report the speed reading result with the highest accuracy. We list the reported FLOP reductions when available. If FLOP reductions are not reported in the original paper, we report the speed increase, which should be considered a lower bound on the FLOP reduction. Note that the state-of-the-art models use different network configurations of the RNN network and training schemes, resulting in different full read and speed read accuracies for the same dataset. To allow a consistent comparison of the *effect* of each speed reading model, we report the accuracy difference between each paper's reported full read (vanilla LSTM) and speed read accuracies.

On all datasets our approach provides either the best or shared best FLOP reduction, except on CBT-CN where LSTM-Jump provides a speed increase of 6.1x (compared to 3.9x for our approach). The second best method with regards to FLOP reduction is Skim-LSTM, and the worst is the Adaptive-LSTM model that implements early stopping when the model is certain of its prediction. Skim-LSTM has an evaluation advantage in this FLOP reduction setting, since the FLOP reduction is directly tied to the difference between the small and large LSTM used by the model, such that an unnecessarily large LSTM will lead to very attractive reductions. Skim-LSTM uses a LSTM size of 200 for Question Answering tasks and a default size of 100 for the rest, whereas the small is tested between 5 to 20. In the case of large skimming percentage, it could be argued that the size of the large LSTM could be reduced without affecting the performance. In contrast, jumping based models are less prone to this evaluation flaw because they cannot carry over information from skipped or jumped words.

Most models perform at least as well as a vanilla LSTM. LSTM-Shuttle provides consistent accuracy improvements, but does so at a noticeable FLOP reduction cost compared to Skim-LSTM and our

| Model | IMDB | | | DBPedia | | | Yelp | | | AG news | | |
|---|---|---|---|---|---|---|---|---|---|---|---|---|
| | Acc | Jump | Read | Acc | Jump | Read | Acc | Jump | Read | Acc | Jump | Read |
| vanilla LSTM | **0.882** | 0% | 100% | 0.972 | 0% | 100% | 0.955 | 0% | 100% | 0.880 | 0% | 100% |
| Structural-Jump-LSTM (ours) | **0.882** | 70.7% | 19.7% | **0.985** | 68.1% | 17.5% | **0.958** | 31.2% | 68.8% | **0.883** | 32.2% | 52.0% |
| Model | SST | | | Rotten Tomatoes | | | CBT-CN | | | CBT-NE | | |
| | Acc | Jump | Read | Acc | Jump | Read | Acc | Jump | Read | Acc | Jump | Read |
| vanilla LSTM | 0.837 | 0% | 100% | 0.787 | 0% | 100% | 0.515 | 0% | 100% | 0.453 | 0% | 100% |
| Structural-Jump-LSTM (ours) | **0.841** | 19.1% | 53.9% | **0.790** | 0.4% | 57.8% | **0.522** | 67.4% | 32.6% | **0.463** | 68.7% | 31.3% |

Table 2: vanilla LSTM refers to a standard LSTM full reading. The columns show the accuracy (Acc), the percentage of text being jumped over, and the total reading percentage.

| Model | IMDB | | DBPedia | | Yelp | | AG news | |
|---|---|---|---|---|---|---|---|---|
| | $\Delta$Acc | FLOP-r | $\Delta$Acc | FLOP-r | $\Delta$Acc | FLOP-r | $\Delta$Acc | FLOP-r |
| Structural-Jump-LSTM (ours) | 0.000 | **6.3x** | **0.013** | **7.0x** | **0.003** | **1.9x** | 0.003 | **2.4x** |
| Skim-LSTM (Seo et al., 2018) | 0.001 | 5.8x | - | - | - | - | 0.001 | 1.4x |
| LSTM-Jump (Yu et al., 2017) | 0.003 | 1.6x* | - | - | - | - | 0.012 | 1.1x* |
| Yu-LSTM (Yu et al., 2018) | 0.005 | 3.4x | 0.002 | 2.3x | 0.002 | 1.4x | 0.001 | 1.7x |
| LSTM-Shuttle (Fu & Ma, 2018) | **0.008** | 2.1x* | - | - | - | - | **0.020** | 1.3x* |
| Adaptive-LSTM (Huang et al., 2017) | - | - | -0.016 | 1.1x* | - | - | -0.012 | 1.1x* |
| Model | SST | | Rotten Tomatoes | | CBT-CN | | CBT-NE | |
| | $\Delta$Acc | FLOP-r | $\Delta$Acc | FLOP-r | $\Delta$Acc | FLOP-r | $\Delta$Acc | FLOP-r |
| Structural-Jump-LSTM (ours) | **0.004** | **2.4x** | 0.003 | **2.1x** | 0.007 | 3.9x | 0.010 | **4.1x** |
| Skim-LSTM (Seo et al., 2018) | 0.000 | **2.4x** | **0.017** | **2.1x** | 0.014 | 1.8x | 0.024 | 3.6x |
| LSTM-Jump (Yu et al., 2017) | - | - | 0.002 | 1.5x* | **0.044** | **6.1x*** | **0.030** | 3.0x* |
| Yu-LSTM (Yu et al., 2018) | - | - | - | - | - | - | - | - |
| LSTM-Shuttle (Fu & Ma, 2018) | - | - | 0.007 | 1.7x* | - | - | 0.019 | 3.0x* |
| Adaptive-LSTM (Huang et al., 2017) | - | - | - | - | - | - | - | - |

Table 3: Comparison of state-of-the-art speed reading models. $\Delta$Acc is the difference between the accuracy of the full read LSTM and the model (the higher the better), and FLOP-r is the FLOP reduction compared to a full read model. A star (*) indicates that the original paper provided only a speed increase, which should be considered a lower bound for FLOP-r.

approach. This can be explained by its ability to make backwards jumps in the text in order to re-read important parts, which is similar to the idea of Yu-LSTM. The largest accuracy improvements appear on CBT-CN and CBT-NE with LSTM-Jump. The performance obtained by reading just the query has been reported to be very similar to using both the query and the 20 sentences (Hill et al., 2016), which could indicate a certain noise level in the data that speed reading models are able to identify and reduce the number of read words between the high information section of the text and the final prediction. LSTM-Jump and LSTM-Shuttle are optimized via maximizing a jumping budget, where only a certain specified number of jumps are allowed to be made, which provides an edge in comparison to the other methods in this setting because prior knowledge about the high information level of the query can be encoded in the budget (cf. Yu et al. (2017) where the best accuracy is obtained using 1 jump for CBT-CN and 5 for CBT-NE). In the setting of speed reading the query is read first to condition the jumping based on the query – this makes the model very likely to prefer jumping shortly after the query is read, to not degrade the LSTM state obtained after reading the query. Overall, budgets can be beneficial if prior information about the document is available, but this is most often not the case for a large set of real world datasets. However, methods based on budgets are in general significantly more rigid, as every document in a collection has the same budget, but the required budget for each document is not necessarily the same.

## 5   CONCLUSION

We presented Structural-Jump-LSTM, a recurrent neural network for speed reading. Structural-Jump-LSTM is inspired by human speed reading, and can skip irrelevant words in important sections, while also jumping past unimportant parts of a text. It uses the dynamically spaced punctuation structure of text to determine whether to jump to the next word, the next sub-sentence separator (,;), next end of sentence (.!?), or to the end of the text. In addition, it allows skipping a word after observing it without updating the state of the RNN. Through an extensive experimental evaluation against all five state-of-the-art baselines, Structural-Jump-LSTM obtains the overall largest reduction in floating point operations, while maintaining the same accuracy or even improving it over a vanilla LSTM model that reads the full text. We contribute the first ever neural speed reading model

that both skips and jumps over dynamically defined chunks of text without loss of effectiveness and with notable gains in efficiency. Future work includes investigating other reward functions, where most of the reward is not awarded in the end, and whether this would improve agent training by having a stronger signal spread throughout the text.

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
