# OpenReview forum: "Neural Speed Reading with Structural-Jump-LSTM"
_ICLR.cc/2019/Conference_

### Official Review · AnonReviewer3 · 2018-10-31
**The paper presents a new speed reading model by combined several existing ideas. The idea is novel and the results are good.**

**Rating:** 7
**Confidence:** 4

**Review:**

The paper presents a novel model for neural speed reading. In this new model, the authors combined several existing ideas in a nice way, namely, the new reader has the ability to skip a word or to jump a sequence of words at once. The reward of the reader is mixed of the final prediction correctness and the amount of text been skipped. The problem is formulated as a reinforcement learning problem. The results compared with the existing techniques on several benchmark datasets show consistently good improvements.

In my view, one important (also a little surprising) finding of the paper is that the reader can make jump choices successfully with the help of punctuations. And, blindly jumping a sequence of words without even lightly read them can still make very good predictions.

The basic idea of the paper, the concepts of skip and jump, and the reinforcement learning formulation are not completely new, but the paper combined them in an effective way. The results show good improvements majorly in FLOPS.

The way of defining state, rewards and value function are not very clear to me. Two value estimates are defined separately for the skip agent and the jump agent. Why not define a common value function for a shared state? Two values will double count the rewards from reading. Also, the state of the jump agent may not capture all available information. For example, how many words until the end of the sentence if you make a jump. Will this make the problem not a MDP?

Overall, this is a good paper.

I read the authors' response. The paper should in its final version add the precise explanation of how the two states interact and how a joint state definition differs from the current one.

---

> ### Author Response · Authors · 2018-11-19
> **Response to AnonReviewer3**
>
> Thank you for your review, questions and suggestions. We address both questions and suggestions below.
>
> Question1: "The basic idea of the paper, the concepts of skip and jump, and the reinforcement learning formulation are not completely new, but the paper combined them in an effective way. The results show good improvements majorly in FLOPS. The way of defining state, rewards and value function are not very clear to me. Two value estimates are defined separately for the skip agent and the jump agent. Why not define a common value function for a shared state? Two values will double count the rewards from reading. [...]"
>
> Answer1: In our model we choose to have a value estimate for each agent, as we posit that reading some high information words can change the state of the LSTM significantly, leading to a different value estimate from the skip agent (which is based on the old LSTM state and the input to the LSTM) and the jump agent (which is based on the updated LSTM state). In principle, if we assume the skip agent can learn how a given word will change the LSTM state, the value estimate from the skip agent could be used for the jump agent, if it is updated to reflect the cost associated with reading the word.
> We have not included in the paper the impact of this on model training and performance explicitly, due to space constraints, but we will investigate it in future work.
>
> Question2: "[...] Also, the state of the jump agent may not capture all available information. For example, how many words until the end of the sentence if you make a jump. Will this make the problem not a MDP?"
>
> Answer2: Whether it is a MDP depends on how we consider the setting when reading the texts. In a streaming setting where each word is continuously arriving, we would not have this information when the decision to jump is made. If we have access to the whole text, we could have access to this information, and our state therefore does not capture all relevant information when making the decision. We have chosen not to use this information, as no other related work uses “future” information when making a decision, but it can potentially give an advantage.
> Similarly to Question1, we have not explicitly extended this discussion in the paper due to space constraints, but we believe it is an interesting idea to try, to see how the policies potentially change when this information is available.

---

### Official Review · AnonReviewer2 · 2018-11-01

**Rating:** 5
**Confidence:** 4

**Review:**

The paper proposes a fast-reading method using skip and jump actions. The paper shows that the proposed method is as accurate as LSTM but uses much less computation.

* pros:
- very fast reading model (?).

* cons:
- although the paper is well written, the jump is not described in details.
- using 'structural-jump' is a little misleading. The model will jump to ".,!" or end of sentence. What is called "structural"? Note that those punctuation marks are not 100% correlated to sentence structure. For example, "He hate fruits such as apples, pears, and oranges." The mode should jump to the end of sentence rather than the first "," when reading "such".
- maybe the authors should say a little bit about the used computation-cost-reduction method. (I.e. in an appendix).

---

> ### Author Response · Authors · 2018-11-19
> **Response to AnonReviewer2**
>
> Thank you for your review, questions and suggestions. We address both questions and suggestions below.
>
> Comment1: As a positive point the reviewer writes: ”very fast reading model (?).”
>
> Answer1: The overall aim of this work was indeed to create a very fast speed-reading model. However, we would also argue that the paper contains multiple contributions in ways of achieving this goal:
> 1) As noted by reviewers 1 and 3, the idea of combining skipping and jumping though a multi-agent architecture has not been done previously and empirically provides state-of-the-art speed-reading results.
> 2) We provide a more stable way of training the speed reading model compared to strong baselines such as LSTM-Jump and LSTM-Shuffle, which both require selecting 3 parameters describing the model constraints from a very large set of possible values. In contrast, because our model makes skip and jump decisions dynamically, we do not have the same tuning of model constraints, and as described in Section 4.1 our parameter tuning is relatively stable independently of the dataset.
>
> Comment2: “although the paper is well written, the jump is not described in details.”
>
> Answer2: In the first paragraph of Section 3 we describe the idea of both the skip and jump agent. The skip agent can skip a single word, thus not updating the LSTM state. If the word is not skipped, the jump agent makes a decision. In practice, both agents output when to read the next word, as the skip agent can decide to ignore the current word and the jump agent can decide to ignore all words until e.g. the next comma. We have now updated the end of Section 3.1 to better describe how the jumping is made based on the sampled action.
>
> Comment3: “using 'structural-jump' is a little misleading. The model will jump to ".,!" or end of sentence. What is called "structural"? Note that those punctuation marks are not 100% correlated to sentence structure. For example, "He hate fruits such as apples, pears, and oranges." The mode should jump to the end of sentence rather than the first "," when reading "such".”
>
> Answer3: Thank you for pointing this out. By "structure" we indeed refer to "punctuation structure". We have now clarified this point throughout the paper.
>
> Comment4: “maybe the authors should say a little bit about the used computation-cost-reduction method. (I.e. in an appendix). “
>
> Answer4: The computation-cost-reduction method is inherent in the speed-reading model, since skipped or jumped words correspond to fewer LSTM update computations. To highlight this point, we have explained explicitly in the end of the first paragraph in Section 3, that the speed up is due to the reduced number of LSTM state update computations.

---

### Official Review · AnonReviewer1 · 2018-11-03
**New incremental work on speed reading with slightly better empirical results**

**Rating:** 7
**Confidence:** 5

**Review:**

The paper proposes a Structural-Jump-LSTM model to speed up machine reading, which is an extension of the previous speed reading models, such as LSTM-Jump, Skim-LSTM and LSTM-Shuffle. The major difference, as claimed by the authors, is that the proposed model has two agents instead of one. One agent decides whether the next input should be fed into the LSTM (skip) and the other determines whether the model should jump to the next punctuation (jump). The sentence-wise jumping makes the jumping more structural than models like LSTM-Jump, while the word-wise skipping operation has a finer skimming decision. The reinforcement learning algorithm in this paper is also different from LSTM-Jump, where LSTM-Jump uses REINFORCE, while this paper applies actor-critic approach.

Empirical studies show that Structural-Jump-LSTM is (slightly) better than state-of-the-art methods in terms of both accuracy and speed over most but few datasets. My feeling is that the proposed model should work much better than the previous models in very long texts, which I suggest the author should try on. Otherwise, the performance gain looks marginal and it is thus questionable whether the complicated modeling is necessary.

I am confused by Figure 1: why are the “yes/no” placed in front of the “skipped”? “Previous LSTM” is confusing as well, which should be “Previous Output/hidden state”.

Minor comment: The LSTM-Jump takes word2vec as the initialization in CBT, while this paper uses GLOVE. I wonder if this results in the performance difference in accuracy. From my experience, GLOVE is usually better than word2vec in most of the tasks. If this effect also applies to CBT, the experiment is not fair.

---

> ### Author Response · Authors · 2018-11-19
> **Response to AnonReviewer1**
>
> Thank you for your review, questions and suggestions. We address both questions and suggestions below.
>
> Question1: “Empirical studies show that Structural-Jump-LSTM is (slightly) better than state-of-the-art methods in terms of both accuracy and speed over most but few datasets. My feeling is that the proposed model should work much better than the previous models in very long texts, which I suggest the author should try on. Otherwise, the performance gain looks marginal and it is thus questionable whether the complicated modeling is necessary.”
>
> Answer1: In our experiments we compare our model against the state of the art using more datasets than any other related work. This large selection of datasets includes texts of very different length. On the datasets with the longest texts (IMDB, CBT-CN, CBT-NE, Yelp) we obtain the largest FLOP reductions on 3 out of 4 of them. IMDB, CBT-CN, CBT-NE are also among the datasets where we obtain the lowest reading percentages (only 19.7% to 32.6%). So our model indeed performs very well on long text. However, we also observe that speed reading is very task -dependent, as one of the datasets with short texts (DBPedia) obtains the lowest reading percentage across all datasets (17.5%).
> Regarding whether “the complicated modeling is necessary”, we note that our model is not notably more complex than related models, as most related models (except Adaptive-LSTM) implement an agent for making speed-reading decisions. In our setting, we use a simple agent for skipping, followed by a potential decision by the structural jumping agent. This allows to effectively combine the benefits of skipping and jumping. Additionally, in comparison to strong models such as LSTM-Jump and LSTM-shuffle, our model makes parameter tuning notably easier: LSTM-Jump and LSTM-Shuffle both require tuning of 3 model constraint parameters describing the jumping behavior, however these vary significantly from dataset to dataset, and are chosen from a large set of values. In contrast, because our model makes skip and jump decisions dynamically, we do not have the same tuning of model constraints, and as described in Section 4.1 our parameter tuning is relatively stable independent of the dataset.
>
> Question2: ”I am confused by Figure 1: why are the “yes/no” placed in front of the “skipped”? “Previous LSTM” is confusing as well, which should be “Previous Output/hidden state”.”
>
> Answer2: The Yes/No refers to which LSTM state and output is used for the next time step – if the word is skipped, then the previous state and output is used, otherwise the current state and output is used. We have now clarified this in the caption of Figure 1. We have also corrected “Previous LSTM” into “Previous Output/hidden state”.
>
> Comment 1: ”Minor comment: The LSTM-Jump takes word2vec as the initialization in CBT, while this paper uses GLOVE. I wonder if this results in the performance difference in accuracy. From my experience, GLOVE is usually better than word2vec in most of the tasks. If this effect also applies to CBT, the experiment is not fair.”
>
> Answer3: This question highlights our reason for reporting accuracy difference, as opposed to absolute values, since the accuracy is dependent on the embedding and (most importantly) model architecture. To answer the question, we have re-run our model on CBT-CN and CBT-NE with the word2vec embedding used in LSTM-Jump and report the results below:
>
> CBT-CN                            Acc.     Jump    Read    FLOP-reduction
> Vanilla LSTM                   0.506
> Structural-Jump-LSTM  0.526   73.0%  26.8%   4.78x
>
> CBT-NE                            Acc.       Jump    Read     FLOP-reduction
> Vanilla LSTM                   0.414
> Structural-Jump-LSTM  0.423    59.4%   33.1%    3.82x
>
> The absolute accuracy scores are lower than when using the GLOVE embedding, however the FLOP reductions and accuracy differences are similar to the GLOVE embedding setting (CBT-CN slightly better and CBT-NE slightly worse). If we replaced our original results on CBT-CN and CBT-NE with these new results, it would not change the ranking of the fastest models on those datasets.
>
> We thank the reviewer for the insightful comments. We hope the above clarifications and paper changes related to Figure 1 sufficiently answer the questions and concerns raised by the reviewer.

---

> > ### Comment · AnonReviewer1 · 2018-12-09
> > **Satisfactory response**
> >
> > I read the author response and feel my questions are well addressed. I will increase the score to champion its acceptance.

---

### Meta-Review · Area_Chair1 · 2018-12-14
**Accept**

**Confidence:** 4
**Recommendation:** Accept (Poster)

**Metareview:**

The authors obtain nice speed improvements by learning to skip and jump over input words when processing text with an LSTM. At some points the reviewers considered the work incremental since similar ideas have already been explored, but at the end two of the reviewers ended up endorsing the paper with strong support.